# An Energy-Efficient Clustering Routing Protocol Based on a High-QoS Node Deployment with an Inter-Cluster Routing Mechanism in WSNs

**DOI:** 10.3390/s19122752

**Published:** 2019-06-19

**Authors:** Kaida Xu, Zhidong Zhao, Yi Luo, Guohua Hui, Liqin Hu

**Affiliations:** 1School of Communication Engineering, Hangzhou Dianzi University, Hangzhou 310018, China; xukaidahdu@gmail.com (K.X.); luoyi@hdu.edu.cn (Y.L.); 2Hangdian Smart City Research Center of Zhejiang Province, Hangzhou Dianzi University, Hangzhou 310018, China; 3College of Electronics and Information, Hangzhou Dianzi University, Hangzhou 310018, China; 4Key Laboratory of Forestry Intelligent Monitoring and Information Technology of Zhejiang Province, School of Information Engineering, Zhejiang A & F University, Linan 311300, China; guohua_hui@zafu.edu.cn; 5Department of Construction Engineering, Zhejiang College of Construction, Hangzhou 311231, China; huliqin98@gmail.com

**Keywords:** wireless sensor network, network coverage, quality of services, inter-cluster routing mechanism, Dijkstra algorithm

## Abstract

Currently, wireless sensor network (WSN) protocols are mainly used to achieve low power consumption of the network, but there are few studies on the quality of services (QoS) of these networks. Coverage can be used as a measure of the WSN’s QoS, which can further reflect the quality of data information. Additionally, the coverage requirements of regional monitoring target points are different in real applications. On this basis, this paper proposes an energy-efficient clustering routing protocol based on a high-QoS node deployment with an inter-cluster routing mechanism (EECRP-HQSND-ICRM) in WSNs. First, this paper proposes formula definitions for information integrity, validity, and redundancy from the coverage rate and introduces a node deployment strategy based on twofold coverage. Then, in order to satisfy the uniformity of the distribution of cluster heads (CHs), the monitoring area is divided into four small areas centered on the base station (BS), and the CHs are selected in the respective cells. Finally, combined with the practical application of the WSN, this paper optimizes the Dijkstra algorithm, including: (1) nonessential paths neglecting considerations, and (2) a simultaneous introduction of end-to-end weights and path weights, achieving the selection of optimal information transmission paths between the CHs. The simulation results show that, compared with the general node deployment strategies, the deployment strategy of the proposed protocol has higher information integrity and validity, as well as lower redundancy. Meanwhile, compared with some classic protocols, this protocol can greatly reduce and balance network energy consumption and extend the network lifetime.

## 1. Introduction

The wireless sensor network (WSN), which has exerted a great influence in the 21st century, combines several advanced technologies, such as wireless communication and sensor technology, to effectively achieve an interaction between human society and the physical world, resulting in a profound impact on promoting social progress. WSNs currently have a wide range of applications in military defense, environmental monitoring, and smart homes. As the basic component of a WSN, the sensor node is generally powered by a dry battery. Once the battery is exhausted, the node is dead and cannot participate in subsequent data operations. If these nodes are placed in harsh environments, it becomes impractical to replace the batteries. Therefore, how to effectively reduce the energy consumption of sensor nodes for long-term operation has become a research hotspot [1].

It was under this context that the WSN clustering routing protocol came into being. The WSN protocol combines a large number of sensor nodes into clusters to form a hierarchical management system from cluster members (CMs) to cluster heads (CHs) to base stations (BS), which can effectively reduce the energy consumption of the network, thus extending the lifetime of the network and achieving more rounds of data iteration.

In recent years, researchers in various countries have proposed a variety of WSN clustering routing protocols. The most classic protocol is LEACH (Low-Energy Adaptive Cluster Hierarchical) [2], which achieves the goal of energy balance by periodically changing the CHs in the area. However, because it adopts a random number mechanism, the suitability of the CH is full of uncertainties.

Hosen and Cho [3] proposed an energy center-based WSN routing mechanism that mainly grades all nodes in the area according to the residual energy of the node and the average distance from the member nodes then selects higher-level nodes to become the CHs. A hierarchical routing protocol based on a k-d tree algorithm was proposed in [4], which uses a spatial partitioned data structure to organize the nodes into clusters. A multi-aware query-driven (MAQD) routing protocol based on a neuro-fuzzy inference system was introduced in [5]. An energy-aware cluster-based CRSN routing protocol (EACRP) was introduced in [6], which considers several factors such as the energy and the dynamic spectrum.

A multi-access WSN routing protocol based on the transmission power control applied to underground coal mines was introduced in [7], which uses the transmission power control algorithm to find the optimal transmission radius and transmission power, and the non-uniform clustering idea is used to optimize the CH selection mechanism. An energy chain-based WSN routing protocol (E-CBCCP) for underwater environments was introduced in [8].

Bozorgi et al. [9] combined static clustering and dynamic clustering, then took the nodes’ residual energy, the energy required to receive the information, and the number of neighbor nodes into consideration to select the CHs. The simulation results showed that, compared with other methods, this method could improve the stability and efficiency of the network. The CH selection mechanism introduced in [10] was divided into three stages: only the advanced node was selected as the CH in Stage 1; all the nodes performed CH selection with the same probability in Stage 2; and the planar topology replaced the clustering topology in Stage 3. The cluster structure optimization problem was divided into multiple sub-optimization problems and solved by multi-objective evolutionary algorithms (MOEAs) in [11]. Combining a new network structure model with the original energy consumption model, a new method to determine the optimal number of clusters was proposed in [12], and through the AGglomerative NESting (AGNES) algorithm, including: (1) introduction of distance variance, (2) the dual-cluster heads (D-CHs) division of the energy balance strategy, and (3) the node dormancy mechanism, it can achieve a reduction in the network energy consumption decay rate and prolong the network lifetime.

In the process of cluster construction, we can also consider the use of principal component analysis [13], fuzzy logic [1], K-means [14], Markov models [15], the introduction of mobile nodes [16,17], etc., to construct a reliable and secure sensor network [18,19,20].

Meanwhile, it is necessary to consider the optimization problem of the information transmission path between the CHs in the clustering process, which must involve the concept of multi-hops [21]. Common path optimization algorithms include the Dijkstra algorithm [22,23,24], the ant colony optimization (ACO) algorithm [25], etc. Through the proper path optimization, the phenomenon of premature death of the CHs and information congestion [26] could be alleviated effectively.

In addition, a large amount of data is generated during the WSN’s environmental monitoring [27]. How to obtain the most complete and effective information possible while minimizing the amount of redundant information is related to the WSN’s quality of service (QoS) [28]. How to build a compliant WSN through a reasonable node deployment [17], [29,30] has also become one of the factors to consider in the protocol. In [31], a clustering technique is proposed that uses the fuzzy c-means clustering method for the formation of clusters and a cluster head is selected based on four parameters: sensor’s location within cluster, location with respect to the fusion center (FC), its signal-to-noise (SNR), and its residual energy.

In real life, a WSN often faces two problems: (1) There are often some monitoring targets in the monitoring area that have key roles. Compared with other points, these targets often have higher coverage requirements. Therefore, for these points we need to arrange more sensor nodes around them to meet their coverage requirements. (2) With the continuous operation of the WSN, the CH far from the BS tends to die prematurely due to the long transmission distance. Therefore, how to build an energy-efficient and reliable information transmission path between the CHs is also worthy of our attention. It is worth noting that the above two problems are still in [12]. The random node deployment strategy in [12] does not guarantee the obtained information’s QoS and thus must not meet the requirement of the QoS for the actual application. In addition, it does not consider the inter-cluster routing mechanism, which will cause some CHs to deplete energy prematurely because of the long transmission distances from the BS. From the perspective of the two aspects above, this paper proposes an energy-efficient clustering routing protocol based on a high-QoS node deployment with an inter-cluster routing mechanism (EECRP-HQSND-ICRM). The basic premise is as follows: For all the monitoring target points with a coverage requirement of two, this paper proposes a series of definition formulas for information integrity, validity, and redundancy and a node deployment strategy based on twofold coverage. Subsequently, in order to satisfy the uniformity of the CH distribution, the BS is divided into four small cells, and the CHs are selected in the respective cells. Finally, combined with the practical application of the WSN, this paper optimizes the Dijkstra algorithm, including: (1) nonessential path neglecting considerations, and (2) a simultaneous introduction of end-to-end weights and path weights, achieving the selection of optimal information transmission paths between the CHs.

The main contributions of this paper include the following:A proposal for calculating the information integrity, validity, and redundancy based on coverage.The determination of a node deployment strategy based on twofold coverage.The construction of the CH selection mechanism through partition based on the uniformity of the CH distribution, the residual energy of the nodes, and the distances from the nodes to the BS.The optimization of Dijkstra to achieve Dijkstra’s computational complexity reduction.

The remainder of this paper is organized as follows: Section 2 introduces a node deployment strategy based on twofold coverage. Section 3 describes the details of the protocol in the paper. The simulation study is conducted in Section 4. Finally, Section 5 summarizes the research and prospects for the future.

## 2. The Node Deployment Strategy Based on 2-Fold Coverage

In this section, we propose a series of calculation formulas for the coverage-based QoS metrics, including information integrity, validity, and redundancy, according to the phenomenon of the different coverage requirements of the monitoring target points in the real area. At the same time, we also determine the node deployment strategy based on 2-fold coverage.

### 2.1. Coverage

As a key factor of the WSN’s QoS, coverage is defined as the proportion of target points that all sensor nodes can monitor to the number of all target points in the area under the 0-1 perception model in this paper, where the 0-1 perception model is a typical idealized model. The target within the perceived radius of the sensor node can be 100% perceived by the node, but the target outside the perceived radius cannot be perceived by the sensor node. Figure 1 shows a typical 0-1 perception model diagram.

The calculation formula for the network coverage is as follows:(1)$$Coverage=\frac{\sum_{s\in S_{area}}T\left(s\right)\Delta s}{\sum_{s\in S_{area}}\Delta s}.$$

In the above formula, *s* represents the monitoring target point, ΔS represents the area occupied by the monitoring target point, *S_area_* represents the monitoring area, and *T*(*s*) represents the coverage condition of the monitoring target point in the area. If it can be monitored by the surrounding sensor nodes, then *T*(*s*) = 1; otherwise, *T*(*s*) = 0.

### 2.2. The Difference in Coverage Requirements

In real-world applications, some key monitoring targets in the area often require more sensor nodes to monitor, that is, their coverage requirements are higher than others. Figure 2 shows a 3-coverage example. If the coverage requirement of the monitoring target is 2, then the information will be redundant. If the coverage requirement of the monitoring target is 3, then the coverage number of the target point meets its requirement. If the coverage requirement of the monitoring target is 4, then a partial loss of valid information occurs.

### 2.3. Coverage-Based QoS Metrics

In the process of defining the coverage-based QoS metrics, this paper is based on the following three assumptions:During each iteration round, the amount of data each sensor node perceives from the environment is the same.The sensory model of the sensor node adopts the 0-1 perception model.The area contains a large number of monitoring target points, and the occupied areas of these points are equal in size.

(1) Information Integrity: Information integrity is the percentage of the active ingredient in the obtained information required for the entire monitoring area. The effective component, *EIG,* of the information refers to information that is not greater than the target’s coverage requirement in the obtained information, and the calculation formula is as follows:(2)$$EIG=\sum_{s\in S_{area}}\min\left(M\left(s\right),J\left(s\right)\right)\Delta s,$$

In the above formula, *M*(*s*) and *J*(*s*) represent the coverage requirement and the actual coverage number of the monitoring target point in the area, respectively.

The information required for the entire monitoring area, *RIF,* refers to the information that satisfies all the coverage requirements, and its calculation formula is as follows:(3)$$RIF=\sum_{s\in S_{area}}M\left(s\right)\Delta s,$$

The calculation formula for information integrity is as follows:(4)$$Integrity=\frac{\sum_{s\in S_{area}}\min\left(M\left(s\right),J\left(s\right)\right)\Delta s}{\sum_{s\in S_{area}}M\left(s\right)\Delta s},$$

If the coverage requirement of all target points is 2, then the above formula can be modified to
(5)$$Integrity=\frac{\sum_{s\in S_{area}}\min\left(2,J\left(s\right)\right)\Delta s}{\sum_{s\in S_{area}}2\Delta s},$$

(2) Information Validity: Information validity is the percentage of the active ingredients in the obtained information. The calculation formula is as follows:(6)$$Validity=\frac{\sum_{s\in S_{area}}\min\left(M\left(s\right),J\left(s\right)\right)\Delta s}{\sum_{s\in S_{area}}J\left(s\right)\Delta s},$$

If the coverage requirement of all target points is 2, then the above formula can be modified to
(7)$$Validity=\frac{\sum_{s\in S_{area}}\min\left(2,J\left(s\right)\right)\Delta s}{\sum_{s\in S_{area}}J\left(s\right)\Delta s},$$

(3) Information Redundancy: Redundancy is the percentage of the redundant ingredients in the obtained information. The calculation formula is as follows:(8)$$Redundancy=\frac{\sum_{s\in S_{area}}\left[J\left(s\right)-min\left(M\left(s\right),J\left(s\right)\right)\right]\Delta s}{\sum_{s\in S_{area}}J\left(s\right)\Delta s}=1-Validity,$$

### 2.4. Node Deployment

From the three coverage-based QoS metrics proposed above, a node deployment strategy needs to improve information integrity and validity, as well as reduce information redundancy as much as possible.

According to existing research, when conducting a 1-fold coverage-based node deployment, if the full coverage condition is satisfied and the number of nodes in the area is minimized, it is necessary to make full use of the coverage disk of each node; that is, the overlapping area between the nodes should be as small as possible. This study found that the effect of constructing a hexagonal honeycomb grid is optimal. Based on this, we propose a 2-fold coverage-based node deployment strategy in a cellular network for the case where the coverage requirement of all monitoring target nodes in the region is 2, as shown in Figure 3.

As shown in Figure 4, the node deployment strategy in this paper is based on a cellular network. The main steps are as follows:(1)A cellular network is built in the monitoring area.(2)The sensor nodes are arranged at the upper left, upper right, and lower middle vertices of the grid. At this point, the 1-fold coverage-based node deployment is completed.(3)The sensor nodes are arranged at the vertices of the center of the grid. At this point, the 2-covered node is deployed.

## 3. The Clustering Protocol

The main steps of the clustering protocol in this paper are as follows:(1)The sensor nodes in the area are deployed according to the 2-fold coverage-based node deployment strategy introduced in Section 2.(2)Centering on the BS, the monitoring area is divided into four small cells, and a certain number of CHs in each cell are selected to meet the uniformity of the CHs distribution.(3)All CHs send broadcasts to the surrounding ordinary nodes, and the latter sequentially add the clusters to which the strongest signals belong according to the strength of the signals received, notifying the corresponding CHs to complete the cluster construction.(4)Combined with practical applications, this paper optimizes the Dijkstra algorithm to achieve the selection of optimal information transmission paths among the CHs.(5)Data are transmitted and energy is updated.

To minimize the total energy consumption and balance the energy consumption of the nodes in the network, we reselect the CHs in the area after each round of data transmission. Additionally, Step 1 is called the node deployment phase of the protocol, Steps 2 and 3 are collectively called the cluster setup phase of the protocol, and Step 4 is called the inter-cluster routing mechanism (ICRM) establishment phase of the protocol. Steps 2, 3, and 4 are collectively called the preparation phase of the protocol. Finally, Step 5 is called the stabilization phase of the protocol.

Before entering the stabilization phase, the CH of each cluster needs to create a corresponding time division multiple access (TDMA) schedule and send a control message named *Schedule_Msg* to its member nodes in the form of (*Node NO.1, Time Slot 1; Node NO.2, Time Slot 2;......*). The time slot allocation of the clustering routing protocol in this paper is provided in Figure 5. A flowchart of the clustering routing protocol in this paper is presented in Figure 6.

### 3.1. CH Selection

In the traditional protocols, while selecting the CHs, the residual energy and position of the node are often neglected, resulting in an uneven final distribution of the selected CHs, among which the CHs with lower residual energy tend to die prematurely.

In Figure 7, the green dot is the CH, and the red dot is the BS. It can be determined that in the process of CH selection, the LEACH protocol will generate a serious phenomenon of an uneven CH distribution, resulting in a relatively large distance between the CH and its member node, which inevitably has an unfavorable effect on the extension of the network lifetime. 

Based on this phenomenon, this paper uses the BS as the center and divides the area into four small cells. Then, based on the idea of CH selection in [31], combining the residual energy of a node with the distance from the node to the BS, we can construct a CH selection factor and select the top 25% of the CH selection factors as the CHs in each of the small cells. The formula for the CH selection factor is as follows:(9)$$S_{CH}\left(E\left(i\right),\;d_{toBS}^{i}\right)=\alpha \cdot Nor\left(E\left(i\right)\right)+\beta \cdot Nor\left(d_{toBS}^{i}\right),$$

In the above formula, α and β represent weight factors, where α+β=1;E(i) represents the residual energy of node *i*; $d_{toBS}^{i}$ represents the distance from node *i* to the BS. Nor() represents the normalization. The larger the value of $S_{CH}$ of node *i*, the more likely it is to be selected as the CH.

Through the mechanism, an appropriate amount of nodes with larger residual energy and shorter distances from the BS are selected as CHs in each small area, which can better ensure that there is a close transmission distance between each CM and the CH of the joined cluster. It avoids the loss of too much residual energy of some nodes due to its large transmission distance. Figure 8 shows a CH distribution diagram of the protocol in this paper. It can be determined that the distribution of the CHs is relatively uniform and can meet the low power consumption of the network.

### 3.2. Energy Consumption Model

This paper adopts the energy consumption model proposed in [32]. In the process of information transmission, the model contains two additional models based on the transmission distance: the free space model and the multipath fading channel model.

$E_{T}(l,d)$ represents the energy consumed by wirelessly transmitting an *l*-bit message. The expression is as follows:(10)$$E_{T}(l,d)=\left\{ \begin{array}{l} l\cdot E_{elec}+l\cdot \varepsilon_{fs}\cdot d^{2},\;d<d_{0}\\ l\cdot E_{elec}+l\cdot \varepsilon_{mp}\cdot d^{4},\;d\geq d_{0} \end{array}\right.,$$

In the above formula, *E_elec_* represents the energy consumed per bit by the transmitter or receiver circuit, *d* represents the transmission distance between the transmitter and the receiver, $\varepsilon_{fs}$ and $\varepsilon_{mp}$ represent the energy factor per bit in the free space model and the multipath fading channel model, respectively, *l* represents the size of the information, and *d*_0_ represents the transmission distance threshold, which is calculated as follows:(11)$$d_{0}=\sqrt{\frac{\varepsilon_{fs}}{\varepsilon_{mp}}}$$
$E_{R}\left(l\right)$ represents the energy consumed by receiving an *l*-bit message. The expression is as follows:(12)$$d_{0}=\sqrt{\frac{\varepsilon_{fs}}{\varepsilon_{mp}}}E_{R}(l)=l\cdot E_{elec},$$

Next we can calculate the energy consumed by the CH and the energy consumed by the CM in a cluster according to the energy consumption model. During the operation of the WSN, the CM is responsible for transmitting the information sensed from the environment to the corresponding CH. In general, the transmission distance $d_{toCH}$ is less than $d_{0}$. Therefore, the corresponding energy consumption $E_{Mem}$ is calculated as follows:(13)$$E_{Mem}=l\cdot E_{elec}+l\cdot \varepsilon_{fs}\cdot d_{toCH}^{2},$$

On the other hand, the CH is also responsible for receiving and fusing the information of the CMs of the cluster and finally transmitting the information after completing the fusion to the BS for decision making. The energy $E_{CH}$ consumed by the CH is divided into three parts: the energy $E_{CH\_R}$ consumed by receiving, the energy $E_{CH\_FU}$ consumed by fusing, and the energy $E_{CH\_T}$ consumed by transmitting. The formulas of the above three are shown as 14, 15, and 16.
(14)$$E_{CH\_R}=k\cdot l\cdot E_{elec},$$
In the above formula, *k* represents the number of CMs in the cluster.
(15)$$E_{CH\_FU}=\left(k+1\right)\cdot l\cdot E_{DA},$$
In the above formula, $E_{DA}$ represents the energy consumption of each bit of data processed by the CH. Since the CH itself can also sense the surrounding environment information, plus the information of the k⋅l size received from the CMs, the total information of the (k+1)⋅l size needs to be processed.
(16)$$E_{CH\_T}=l\cdot E_{elec}+\left\{ \begin{array}{l} l\cdot \varepsilon_{fs}\cdot d_{toBS}^{2},\;d_{toBS}<d_{0}\\ l\cdot \varepsilon_{mp}\cdot d_{toBS}^{4},\;d_{toBS}\geq d_{0} \end{array}\right.,$$
In the above formula, $d_{toBS}^{}$ represents the distance between the CH and the BS. After information fusion, the original information of the (k+1)⋅l size is compressed into the size of l.

Therefore, the energy $E_{CH}$ consumed by the CH is as follows:(17)$$E_{CH}=E_{CH\_R}+E_{CH\_FU}+E_{CH\_T}=k\cdot l\cdot E_{elec}+\left(k+1\right)\cdot l\cdot E_{DA}+l\cdot E_{elec}+\left\{ \begin{array}{l} l\cdot \varepsilon_{fs}\cdot d_{toBS}^{2},\;d_{toBS}<d_{0}\\ l\cdot \varepsilon_{mp}\cdot d_{toBS}^{4},\;d_{toBS}\geq d_{0} \end{array}\right.,$$

### 3.3. The Dijkstra Algorithm

The Dijkstra algorithm is a routing algorithm proposed by the Dutch computer scientist Dijkstra in 1959 to solve the shortest path optimization problem between the source point and the remaining points in the directed graph. The algorithm takes a greedy strategy, and its main steps are as follows:(1)Construction of an initial weight matrix(2)Updating the weight

Suppose there are *n* vertices in the region, and the set of vertices is *T* = {1,2,...,*n*}. In Step 1, we have *n*^2^ initial weights, and a two-dimensional matrix *W* is used to store the *n^2^* initial weights, where *W*(*i,j*) represents the weight of point *i* to point *j*. It is only if point *i* and point *j* can directly convey information to one another that *W*(*i,j*) = *W*(*j,i*). Otherwise, only point *i* can transmit information directly to point *j*, and point *j* cannot directly transmit information to point *i*, so *W*(*i,j*) is a finite positive number, and *W*(*j,i*) has a value of positive infinity. For *i* = 1,2,…,*n*, *W*(*i,i*) = 0.

In Step 2, we first determine the source point *s*, then establish the vertex set *P* of the known shortest paths and the vertex set *Q* of the unknown shortest paths. Now *P* = {*s*}, and *Q* = *T* − *P*. Here, we declare an array *dis* to store the initial weight of the source point *s* to the remaining points. The representation of *dis* is shown in Figure 9.

Then, we can find the vertex *x* with the smallest weight from itself to s from *Q*, then remove *x* from *Q*, add it to *P,* and compare whether the sum of the weights from point *s* to point *x* to one of the other points *i* is smaller than the weight of point *s* to point *i* directly. If yes, then *W*(*s,i*) = *W*(*s,x*) + *W*(*x,i*), and we replace the corresponding value *W*(*s,i*) in *dis*. Then, we can find the vertex *y* with the smallest weight from itself to *s* from *Q*, remove *y* from *Q,* and add it to *P*, repeating the previous steps until *P* contains all the vertexes in the graph.

The detail procedure of the Dijkstra algorithm is given by the pseudocode in Algorithm 1.

**Algorithm 1.** Dijkstra Algorithm
**Inputs:**  (1) *n*^2^ weights      (2) vertex set, *T***Result:** an array of weights from source point to remaining points, *dis*1: for *i* = 1 → *n*2:  for *j* = 1 → *n*3:   *W*(*i*, *j*) = weight from Point *i* to Point *j*;4:  end for5: end for6: determine the source point in the region, *s*;7: *P* = {*s*}, *Q* = *T* − *P*;8: *dis* = [*W*(*s*, 1), *W*(*s*, 2), …, *W*(*s*, *n*)];9: while (*Q* ~= *Ø*)10:  find the vertex *x* with the smallest weight from the vertex *s* from *Q*;11:  *P* = *P* + {*x*}, *Q* = *Q* - {*x*};12:  for *i* = 1 → *n*13:   if *i* is in *Q*14:    if *W*(*s*, *i*) > *W*(*s*, *x*) + *W*(*x*, *i*)15:     *W*(*s*, *i*) = *W*(*s*, *x*) + *W*(*x*, *i*);16:     update the corresponding value, *W*(*s*, *i*) in *dis*;17:    end if18:   end if19:  end for20: end while

### 3.4. Inter-Cluster Routing Mechanism

In this section, we optimize the Dijkstra algorithm: when defining the weights, we ignore the consideration of the nonessential paths, thus reducing the computational complexity and achieving the selection of optimal information transmission paths among the CHs.

(1) The Definition of Weights: First, based on the energy consumption of the nodes in the network during data transmission, we propose formulas for calculating the weight between the end-to-end and the path weight.

The calculation formula for the weight between the end-to-end is as shown in (18):(18)$$W\left(i,j\right)=E_{T}\left(l,d\left(i,j\right)\right)=\left\{ \begin{array}{l} l\cdot E_{elec}+l\cdot \varepsilon_{fs}\cdot d{\left(i,j\right)}^{2},\;d\left(i,j\right)<d_{0}\\ l\cdot E_{elec}+l\cdot \varepsilon_{mp}\cdot d{\left(i,j\right)}^{4},\;d\left(i,j\right)\geq d_{0} \end{array}\right.$$
In the above formula, *d*(*i,j*) represents the distance between node *i* and node *j*, and the weight between the two is directly represented by the required energy consumption for transmitting information between the two.

The calculation formula for the path weight is as shown in (19):(19)$$W\left(Path\left(M_{1},M_{n+1}\right)\right)=\frac{1}{n}\cdot \sum_{i=1}^{n}W(M_{i},M_{i+1}),$$
In the above formula, we can obtain the weight of the path $W\left(Path\left(M_{1},M_{n+1}\right)\right)$, in which the information starting point is node *M*_1_, which passes through *M*_2_, *M*_3_... in order and finally reaches node *M_n +_*
_1_.

Because in the WSN applications information ultimately needs to be transmitted to the BS, the path is the information transmission path from a CH to the BS in the area. Thus, (19) can be modified to:(20)$$W\left(Path\left(M_{1},BS\right)\right)=\frac{1}{n}\cdot \left(\sum_{i=1}^{n-1}W(M_{i},M_{i+1})+W\left(M_{n},BS\right)\right),$$

(2) The Neglect Consideration of Non-Essential Paths: As described in Section 3.4, the Dijkstra algorithm mainly selects the optimal path by focusing on the starting point and expanding the outer layer. However, when the optimal path traverses the set *Q*, it often generates some unnecessary weight calculations. As an example of a WSN, as shown in Figure 10, it is obvious that the distance *d*(*i,x*) from node *i* to node *x* is larger than the distance *d*(*i,BS*) from node *i* to the BS. If node *x* acts as the relay node of node *i*, unnecessary energy consumption will be generated. When the path of node *i* to the BS is selected by the Dijkstra algorithm, the BS serves as the source point s. When the BS selects the node with the smallest weight from itself, once *x* is selected, the values of *W*(*BS,i*) and *W*(*BS,x*) + *W*(*x,i*) are inevitably compared. As mentioned above, *x* must not act as the relay node between node *i* and the BS, which inevitably causes redundant calculations.

Based on this, we could ignore this situation in the process of optimizing Dijkstra, thus reducing the computational complexity. In addition, the Dijkstra algorithm selects the node *x* closest to the source point *s* from the set *Q* as a multi-hops relay node, and then whether other nodes in the set *Q* have a better multi-hops effect through the node *x* is determined, thereby continuously updating the weights and selecting the optimal information transmission paths. From another point of view, in the process of optimizing Dijkstra, we order all the CHs according to distance from the BS to form an ordered set of CHs, then we traverse the CHs, and each CH is respectively judged with the previous CHs by the path weights until all the CHs in the area find the optimal information transmission paths to the BS.

The detailed procedure of the Optimized Dijkstra algorithm in the WSN is given by the pseudocode in Algorithm 2.

**Algorithm 2.** Optimized Dijkstra Algorithm in WSN
**Inputs:** (1) CHs set, *T*    (2) the coordinate of BS, (*BS.x*, *BS.y*)**Result:** (1) the set of the next hop CH of each CH, *NH*    (2) path set from each CH to BS, *PA*
    (3) an array of path weights from remaining nodes to BS, *dis*1: for *i* = 1 → *n*2:  for *j* = 1 → *n*3:   *W*(*i*,*j*) = weight from Node *i* to Node *j* according to (14);4:  end for5: end for6: dis = [*W(1, BS)*, *W(2, BS)*,…, *W(n, BS)*]^T^, *NH*(1) = BS, *PA*(1) = {BS};7: sort all CHs in *T* in order of the distances from the BS to form an ordered set, *CHs*;8: initialize an array of the hop number for each CH, *HP* = [1,1,…,1]_n_;9: for *i* = 2 → length(*CHs*)10:  for *j* = 1 → *i* − 111:   if *d*(*i*, *j*) < *d*(*i*, *BS*) 12:    if *W*(*Path*(*j*, *BS*))**HP*(*j*) + *W*(*i*, *j*) < *W*(*i*, *BS*)*(*HP*(*j*)+1)13:     *W*(*i*, *BS*) = [*W*(*Path*(*j*, *BS*))**HP*(*j*) + *W*(*i*, *j*)]/(*HP*(*j*)+1);14:     *HP*(*i*) = *HP*(*j*) + 1, *NH*(*i*) = *CHs*(*j*), *PA*(*i*) = *PA*(*j*) + {*CHs*(*i*)};15:     update the corresponding value, *W*(*i*,*BS*) in *dis*;16:    end if17:   end if18:  end for19: end for

## 4. Simulation Results

In this section, we evaluate the proposed protocol by simulation using Spyder (Python 3.6) on a desktop PC (Lenovo, made in Beijing, China) with Intel(R) Core (TM) i3-4170 CPU @ 3.70 GHz, 8GB RAM. The BS of this paper is located at coordinates (50, 45), and the area is a square of 100 m × 90 m. The specific simulation parameters are shown in Table 1. (Note that J in this paper stands for Joule, which is a unit of energy.) 

We mainly compare the node deployment strategy proposed in this paper with the random node deployment strategy, the uniform node deployment strategy, and the 1-fold coverage-based node deployment strategy in a cellular network in terms of coverage, information integrity, information validity, and information redundancy. At the same time, this paper selects three classic protocols, including LEACH, Distribute Energy-Efficient Clustering (DEEC), and Group-Based Sensor Network (GSEN). Then, the protocol of this paper (EECRP-HQSND-ICRM) is compared with the three protocols and the protocol without the inter-cluster routing mechanism (ICRM) of this paper (EECRP-HQSND) to compare the network lifetime, network coverage, information integrity, network energy consumption, the iteration round at which the WSN starts to decline, and the network throughput to reflect the advantages of the protocol in this paper, and especially the feasibility of the ICRM. 

### 4.1. Comparison of the QoS of the Node Deployment Strategies

In this section, we mainly compare the node deployment strategy proposed in this paper with the random node deployment strategy, the uniform node deployment strategy, and the 1-fold coverage-based node deployment in a cellular network in terms of coverage, information integrity, information validity, and information redundancy to reflect the advantages of the node deployment strategy proposed in this paper.

#### 4.1.1. Node Distribution of the Node Deployment Strategies

Figure 11, Figure 12, Figure 13 and Figure 14 show the node distribution diagrams of the random node deployment strategy, the uniform node deployment strategy, the 1-fold coverage-based node deployment strategy in the cellular network, and the node deployment strategy of this paper. The blue point represents the sensor node, the red point represents the BS, each black circle represents the sensing range of each sensor node, and the red square is the monitoring area. It can be determined that in order to satisfy the coverage of the WSN, some of the nodes in Figure 13 and Figure 14 are located outside of the area.

#### 4.1.2. QoS of the Node Deployment Strategies

According to the calculation formulas of the coverage-based QoS metrics introduced in Section 2.3, we compare the coverage, the information integrity, the information validity, the information redundancy, and the number of sensor nodes of the four node deployment strategies above. The specific results are shown in Table 2.

As can be seen from Table 2, in terms of coverage, the random node deployment strategy only reaches 94.233%, while the other three deterministic node deployment strategies reach 100%. In terms of information validity and redundancy, we find that the 1-fold coverage-based node deployment strategy in a cellular network achieves the best effect; that is, this strategy can make the obtained information have a high validity and a low redundancy, which is the main reason why it is widely used in the case of 1-fold coverage requirements. However, the strategy’s information integrity under the 2-coverage requirement is only 59.833%, which obviously cannot meet the basic requirements of the project. In terms of information integrity, the uniform node deployment strategy and the node deployment strategy of this paper both reach 100%. However, compared with the 63.134% information validity, 36.866% information redundancy, and the 110 nodes required by the uniform node deployment strategy, the node deployment strategy in this paper can achieve 83.480% information validity and 16.520% information redundancy and only requires 91 sensor nodes. Obviously, the node deployment strategy in this paper is more advantageous than the uniform node deployment strategy.

In summary, in the four node deployment strategies, the node deployment strategy proposed in this paper has a superior QoS and can meet the basic requirements of the project.

### 4.2. Comparison of the Network Lifetime

As a visual evaluation standard reflecting the performance of clustering routing protocols, the network lifetime refers to the live nodes in the network with the iteration rounds. The steady lifetime refers to the number of rounds until the first node death. In a steady lifetime, the nodes in the network are full of energy and can perform large-scale data communication. This lifetime is the most vigorous moment of the WSN’s vitality. As shown in Figure 15, the steady lifetimes of the LEACH, DEEC, and GSEN are 1061, 1135, and 1163 rounds, respectively, and the steady lifetime of the EECRP-HQSND is 1331, which further exceeds LEACH, DEEC, and GSEN by 25.45%, 17.27%, and 14.45%, respectively. In addition, through a reasonable ICRM, the EECRP-HQSND-ICRM extends the steady lifetime by 33 rounds compared to the EECRP-HQSND. With the iteration rounds, there are few nodes in LEACH and DEEC around the 1300th round. At this time, there is no node death in EECRP-HQSND or EECRP-HQSND-ICRM. After around the 1356th round, the number of nodes in EECRP-HQSND is almost 0, but EECRP-HQSND-ICRM still maintains 91. So, we find that ICRM can reduce the network energy consumption well. In addition, in terms of the downward trend of the curves of the live nodes, it is obvious that compared with LEACH, DEEC, and GSEN, after the death of the first node, the slopes of EECRP-HQSND and EECRP-HQSND-ICRM change the fastest, and the slopes can almost reach negative infinity. Thus we find that compared with the traditional CH selection mechanisms, CH selection through partition based on the uniformity of the CH distribution in this paper can balance the network energy consumption well.

### 4.3. Comparison of the Network Coverage

The network coverage reflects the proportion of monitored targets covered in the area out of the total number, reflecting the network’s monitoring capability. A decline in the network coverage means that the monitoring capability of the network begins to decline. As shown in Figure 16, among the five protocols, the decline of the network monitoring capability under LEACH is the earliest, followed by DEEC and GSEN, while EECRP-HQSND begins to experience a decline in monitoring capacity in the 1333rd round. Through an effective ICRM, the EECRP-HQSND-ICRM delays the monitoring capability decline to the 1367th round, thus ensuring that the network can perform more rounds of monitoring during this period to achieve higher monitoring requirements. Meantime, with a reasonable CH selection mechanism through partition based on the uniformity of the CH distribution, compared with LEACH, DEEC, and GSEN, the round when the network coverage of EECRP-HQSND and EECRP-HQSND-ICRM begins to decline can be greatly delayed.

### 4.4. Comparison of the Information Integrity

Information integrity reflects the proportion of information required to meet the monitored area as a percentage of the total required information. The decline in information integrity, to a certain extent, means that the obtained information has become less desirable. As shown in Figure 17, similar to the network lifetime and coverage, the information integrity of LEACH is the first to decline among the five protocols, followed by the DEEC and the GSEN, while the EECRP-HQSND only experienced a decline in information integrity in the 1332nd round. Finally, the ICRM in this paper can greatly reduce the CHs’ energy load, especially for the CHs that are remotely located, resulting in the EECRP-HQSND-ICRM postponing the decline of information integrity to the 1365th round, thus ensuring that the network can obtain more complete data during this period.

### 4.5. Comparison of the Round at Which the WSN Begins to Decline

In this section, we will analyze the network’s decline from three aspects: the number of rounds until the death of the first node, the number of rounds leading to the decline in network coverage, and the number of rounds leading to the decline in information integrity. Combined with the contents of Section 4.2, Section 4.3 and Section 4.4, our results are shown in Table 3 and Figure 18. It can be seen that compared with the traditional protocols, including LEACH, DEEC, and GSEN, the CH selection mechanism of partition based on the uniformity of the CH distribution proposed in this paper can balance the network energy consumption well, thus greatly delaying the number rounds until the death of the first node, the number of rounds leading to the decline in network coverage, and the number of rounds leading to the decline in information integrity. Then, by constructing the optimal path from each CH to the BS, the ICRM proposed in this paper can further reduce the network energy consumption, thus optimizing the above three aspects.

### 4.6. Comparison of the Network Energy Consumption

In the analysis of network energy consumption, this paper mainly considers two aspects: the network energy consumption size and the network energy consumption uniformity. The index of the network energy consumption uniformity is mainly the range and variance of the residual energy of the nodes in the network.

#### 4.6.1. Comparison of Network Energy Consumption 

In this paper, the BS is centered, and the area is divided into 4 cells. Then, we select the top 25% of nodes as the CHs in each cell and perform the establishment of an ICRM through the optimized Dijkstra algorithm to reduce network energy consumption and balance the energy consumption of the network nodes. As shown in Figure 19, compared with LEACH and DEEC, the GSEN has an advantage in reducing energy consumption. Compared to these three, the EECRP-HQSND effectively reduces the energy consumption of the network, which proves the effectiveness of the CH selection mechanism in this paper. In addition, EECRP-HQSND-ICRM further reduces the network energy consumption compared to EECRP-HQSND, reflecting the feasibility of the ICRM.

#### 4.6.2. Comparison of Network Energy Consumption Uniformity

It can be seen from Figure 20 and Figure 21 that the protocol can effectively balance the energy consumption of nodes in the network compared to LEACH, DEEC, and GSEN. 

In the network iteration, the maximum energy range of LEACH reaches approximately 0.13 J, that of DEEC and GSEN both reach approximately 0.10 J, that of EECRP-HQSND reaches approximately 0.028 J, and that of EECRP-HQSND-ICRM reaches a maximum of 0.01564 J. 

As for the energy variance, EECRP-HQSND EECRP-HQSND-ICRM achieve variances of 0–0.000036. The maximum variance of LEACH reaches 0.0010, that of DEEC reaches 0.0003, and that of GSEN reaches 0.0004.

Based on the comprehensive energy range and variance, it can be determined that the protocol in this paper can achieve a good balance of energy consumption among the nodes in the network.

### 4.7. Comparison of the Network Throughput

The network throughput is an important indicator that fundamentally reflects the performance of a protocol. The network throughput refers to the number of packets in the network that are ultimately sent to the BS. The CMs transmit the sensed information to the CH in the form of packets, and the CH fuses this information with that sensed by itself and finally sends the information to the BS in the form of packets. As the nodes in the network die continuously, the number of CHs in the network decreases, and the success rate of transmitting packets to the BS also decreases. Therefore, the number of data packets transmitted in each round will gradually decrease. Until the last node dies, the network throughput reaches a maximum value, which is just the final network throughput.

As seen from Figure 22, the final network throughput of LEACH is 11005, that of DEEC is 13901, and that of GSEN is 16239. Compared with these three, the EECRP-HQSND increases the final network throughput by 167.53%, 111.80%, and 81.30%, respectively. With an effective ICRM, the EECRP-HQSND-ICRM increases the final network throughput by 856 compared to EECRP-HQSND.

## 5. Conclusions

Aimed at the different coverage requirements of regional monitoring target points in real-world applications, this paper proposes a node deployment strategy based on twofold coverage for all coverage requirements of two in the region. Then, from the network coverage, this paper proposes a series of definition formulas for information integrity, validity, and redundancy. In the CH selection, this paper fully considers the uniformity of the CH distribution and proposes a strategy of dividing the monitoring area into four small cells with the BS as the center and then selecting the CHs in their respective cells. Finally, combined with the actual requirements of the WSN, this paper optimizes the Dijkstra algorithm, including (1) nonessential paths neglecting considerations and (2) a simultaneous introduction of end-to-end weights and path weights. Then, the optimized Dijkstra algorithm is applied to the establishment of the ICRM. 

The simulation results show that, compared with the random node distribution strategy, the uniform node distribution strategy, and the onefold coverage-based node deployment strategy in a cellular network, the node deployment strategy in this paper achieves relatively high coverage, information integrity, and validity, as well as a relatively low information redundancy. At the same time, compared with LEACH, DEEC, and GSEN, EECRP-HQSND-ICRM can reduce and balance network energy consumption, thus extending the network lifetime and improving the network throughput. It is noted that the CH selection mechanism of partition based on the uniformity of the CH distribution in this paper is mainly responsible for selecting CHs with large residual energy and uniform distribution, thereby effectively balancing network energy consumption and delaying the number of rounds until the network performance begins to decline. The ICRM is mainly responsible for finding the optimal transmission path from each CH to the BS in the network and preventing some CHs from dying early because of the long transmission distances from the BS, thereby further reducing network energy consumption and realizing more rounds of network data iterations, which has great significance for engineering. However, there are some shortcomings in our protocol:

First, we used the 0-1 perception model for node deployment and QoS formula derivation. In actual situations, the perception of nodes is very complicated, and it is much less ideal than the 0-1 perception model. Therefore, we may later consider a node deployment scheme and QoS formulas based on a more complex perceptual model.

Finally, the protocol is applicable only to 2D scenarios. Typical 3D scenarios, such as underground coal mines, underground pipe corridors, and indoor homes require the WSN to be arranged in three dimensions. Therefore, in the future, we will consider proposing a clustering routing protocol suitable for 3D scenarios based on this protocol. 

## Figures and Tables

**Figure 1 sensors-19-02752-f001:**
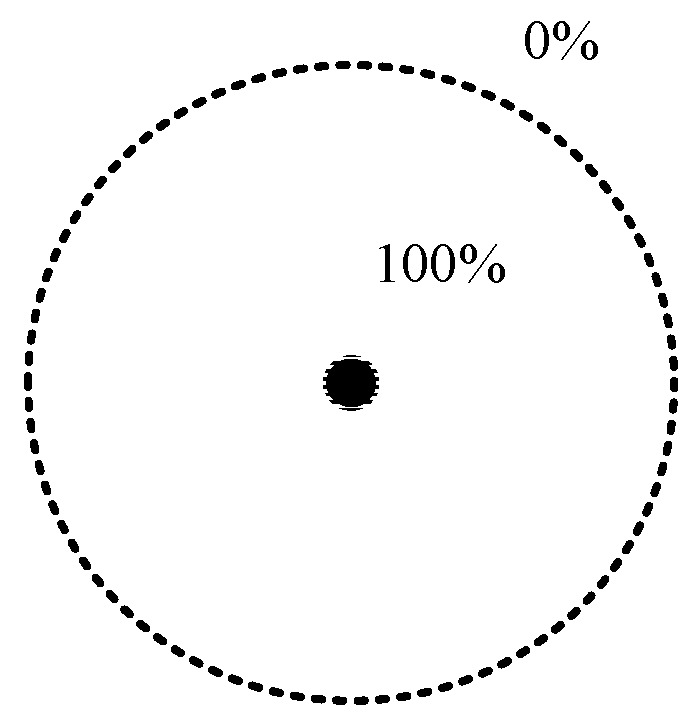
A typical 0-1 perception model diagram.

**Figure 2 sensors-19-02752-f002:**
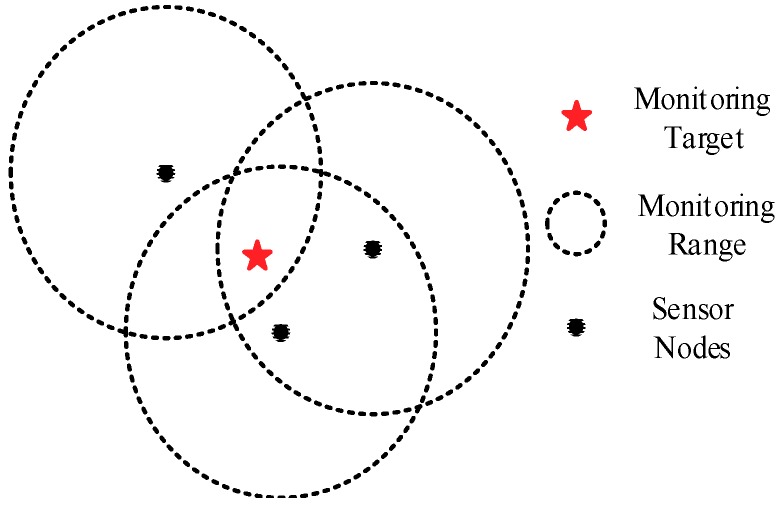
Threefold coverage.

**Figure 3 sensors-19-02752-f003:**
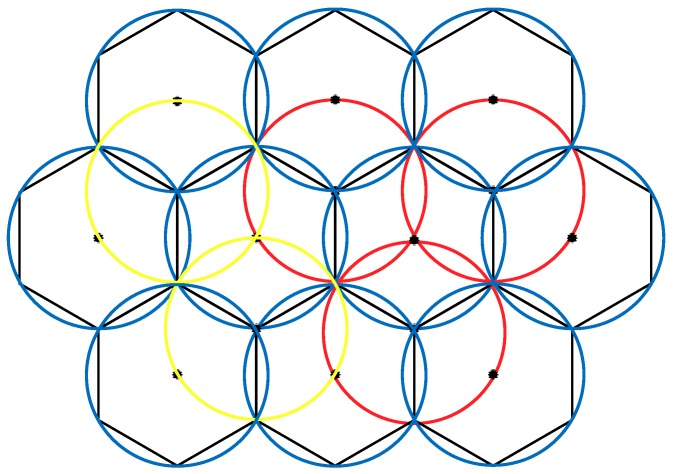
A schematic diagram of a 2-fold coverage-based node deployment strategy in a cellular network.

**Figure 4 sensors-19-02752-f004:**
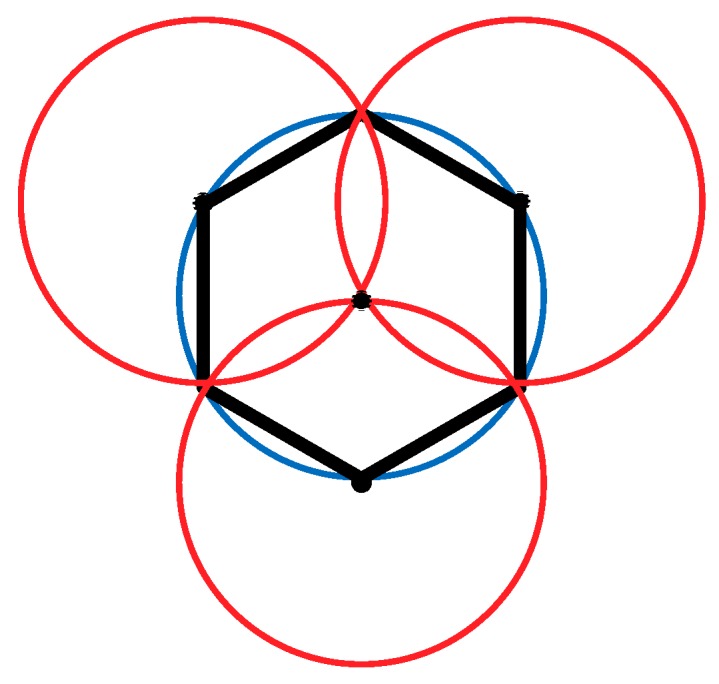
A schematic diagram of the node deployment for a single cellular grid.

**Figure 5 sensors-19-02752-f005:**
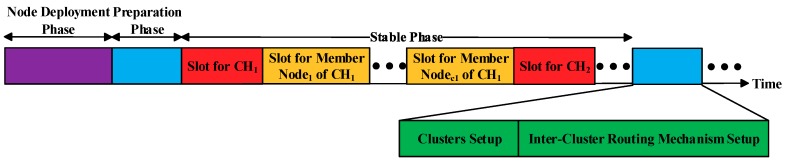
The time slot allocation of the clustering protocol. CH—cluster head.

**Figure 6 sensors-19-02752-f006:**
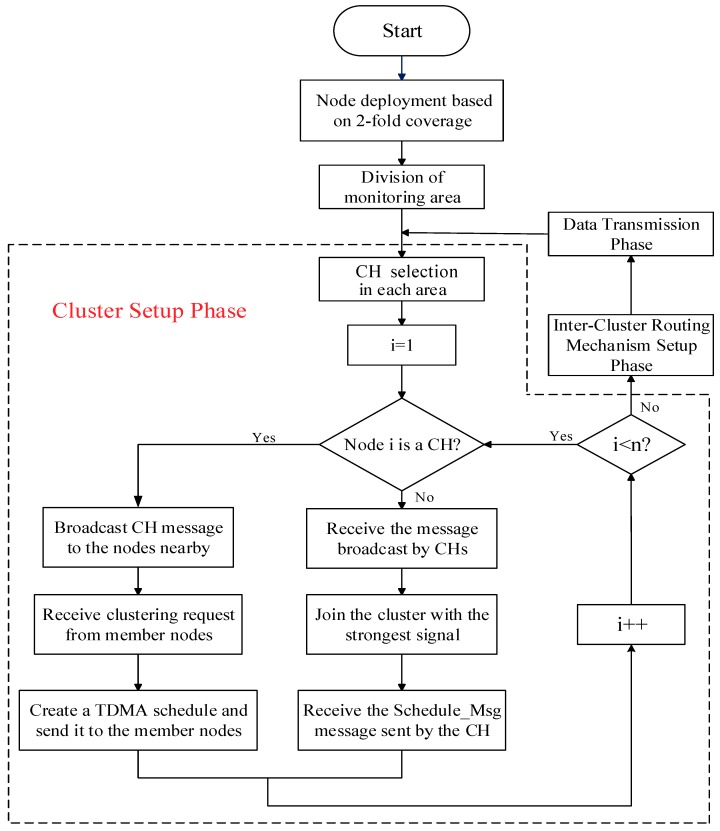
Flowchart of the clustering protocol, including node deployment, CHs selection, cluster setup, and data transmission.

**Figure 7 sensors-19-02752-f007:**
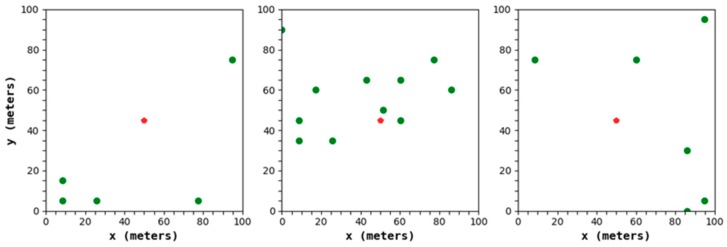
The three rounds of uneven CH distributions generated by the LEACH (Low-Energy Adaptive Cluster Hierarchical) protocol.

**Figure 8 sensors-19-02752-f008:**
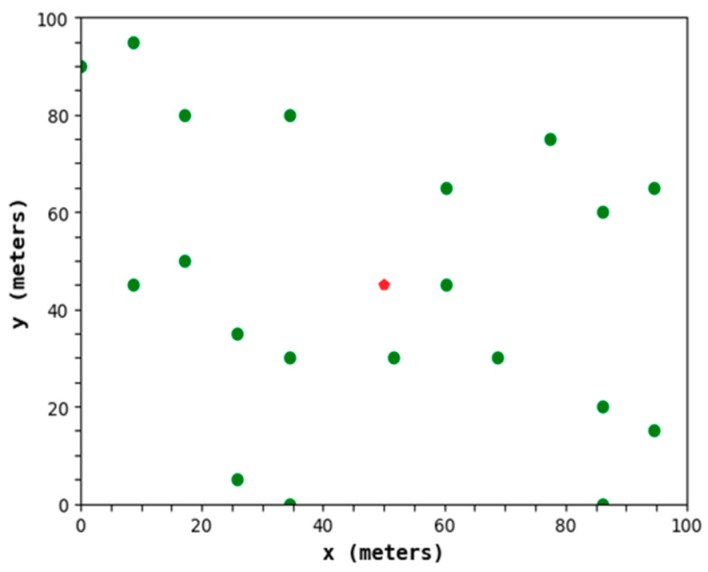
A CH distribution diagram of the protocol in this paper.

**Figure 9 sensors-19-02752-f009:**
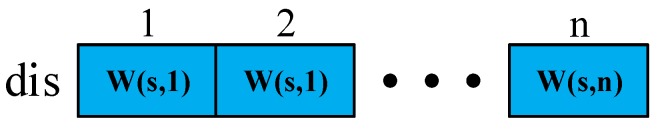
The representation of *dis*.

**Figure 10 sensors-19-02752-f010:**
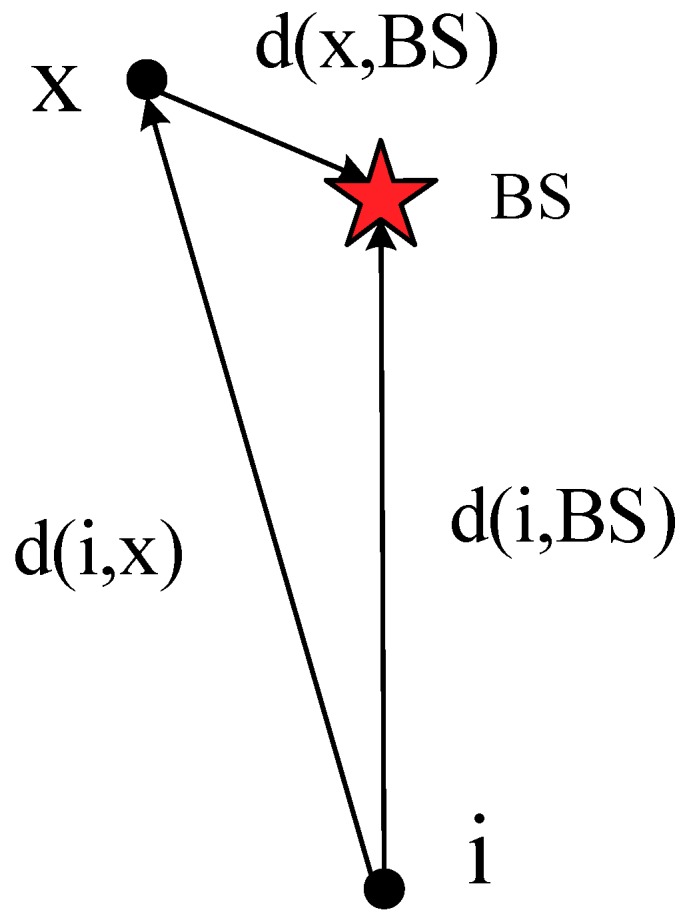
Schematic diagram of information transmission between nodes. BS—base station.

**Figure 11 sensors-19-02752-f011:**
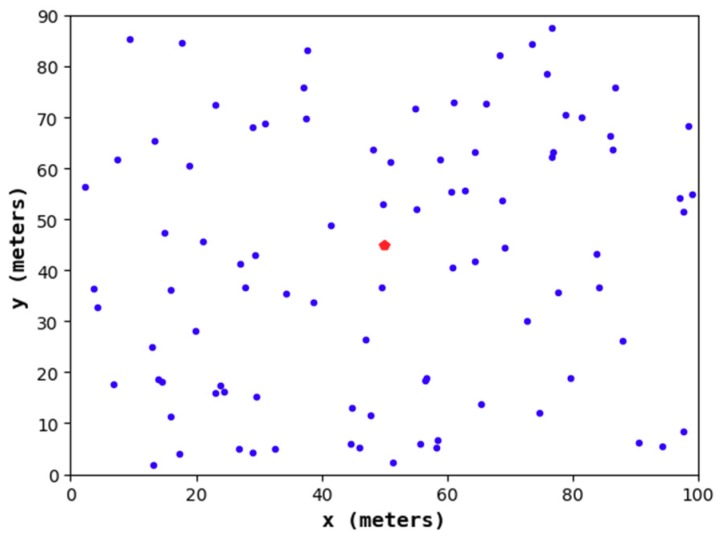
The node distribution diagram of the random node deployment strategy.

**Figure 12 sensors-19-02752-f012:**
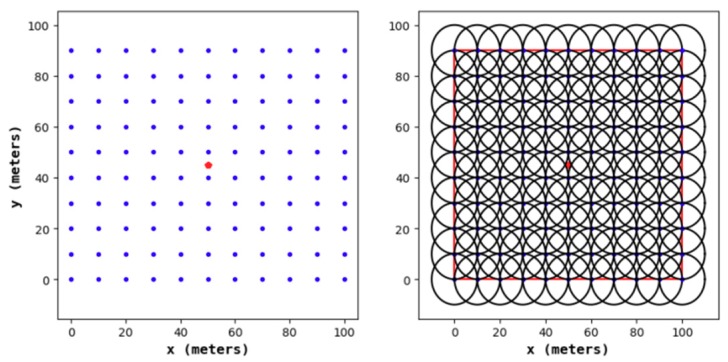
The node distribution diagram of the uniform node deployment strategy.

**Figure 13 sensors-19-02752-f013:**
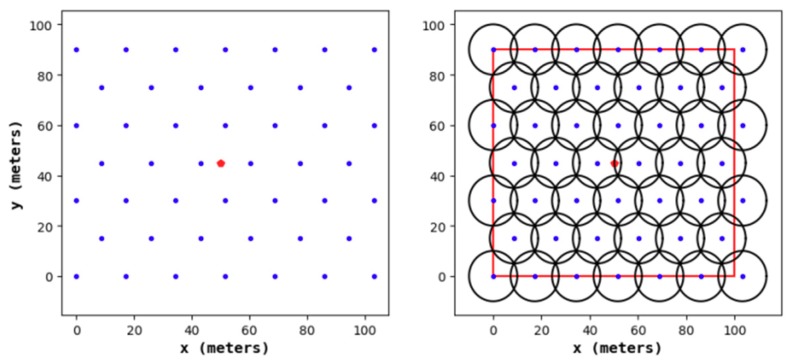
The node distribution diagram of the 1-fold coverage-based node deployment strategy in a cellular network.

**Figure 14 sensors-19-02752-f014:**
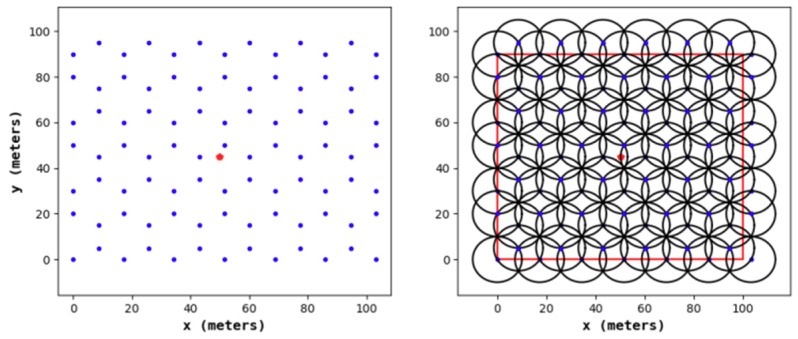
The node distribution diagram of the node deployment strategy proposed in this paper.

**Figure 15 sensors-19-02752-f015:**
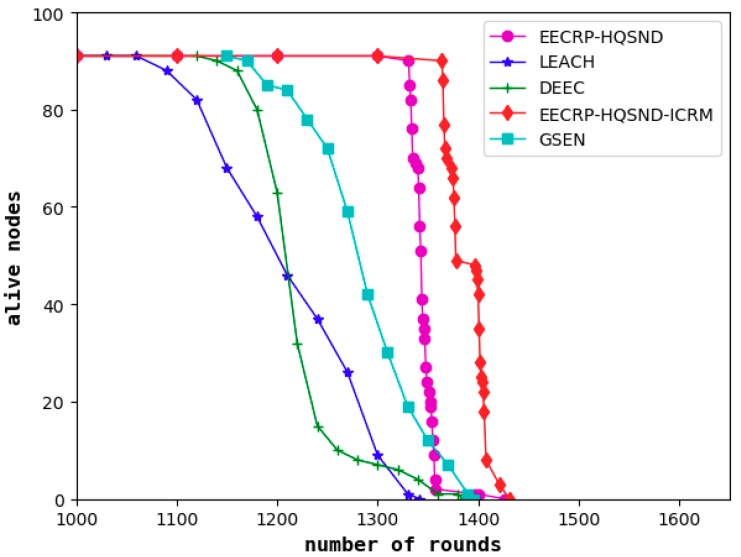
The number of live nodes varies with the iteration round. DEEC— Distribute Energy-Efficient Clustering; EECRP-HQSND—energy-efficient clustering routing protocol based on a high-QoS node deployment; EECRP-HQSND-ICRM—energy-efficient clustering routing protocol based on a high-QoS node deployment with an inter-cluster routing mechanism; GSEN— and Group-Based Sensor Network; LEACH—Low-Energy Adaptive Cluster Hierarchical.

**Figure 16 sensors-19-02752-f016:**
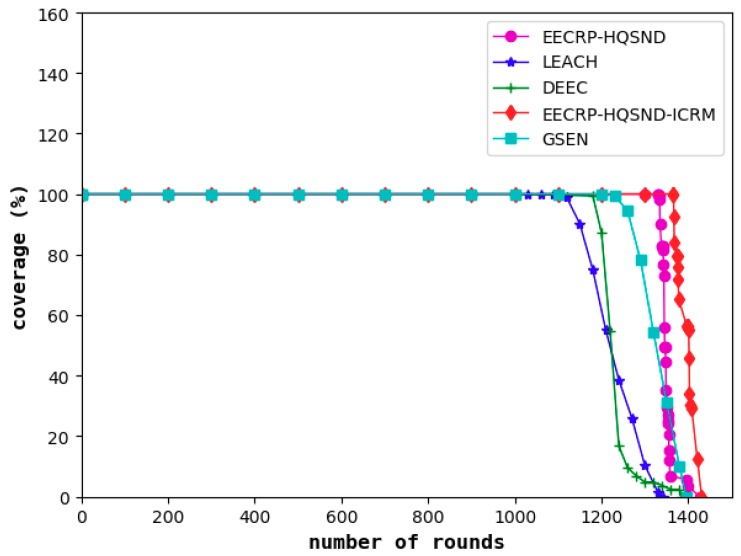
The network coverage varies with the iteration round.

**Figure 17 sensors-19-02752-f017:**
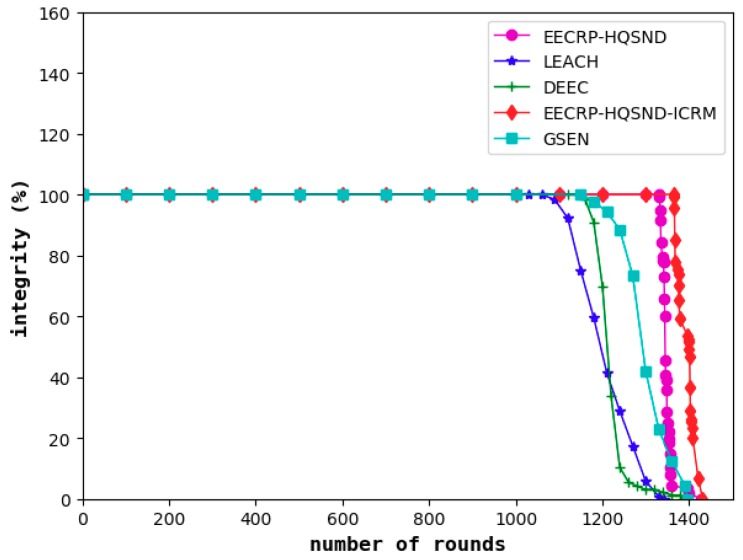
The information integrity varies with the iteration round.

**Figure 18 sensors-19-02752-f018:**
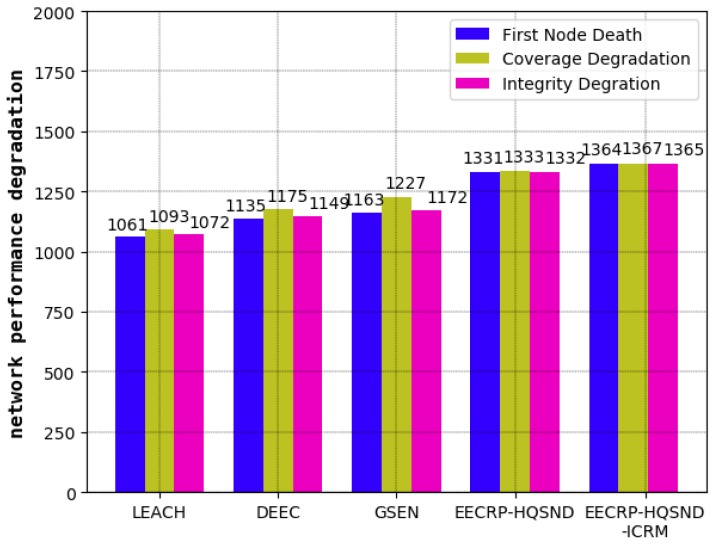
Comparison of the number of rounds leading to decline in the network for each protocol.

**Figure 19 sensors-19-02752-f019:**
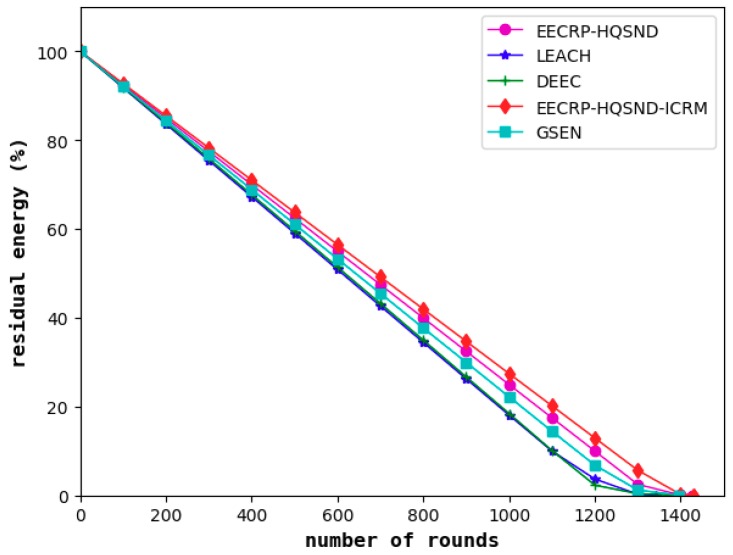
The network residual energy varies with the iteration round.

**Figure 20 sensors-19-02752-f020:**
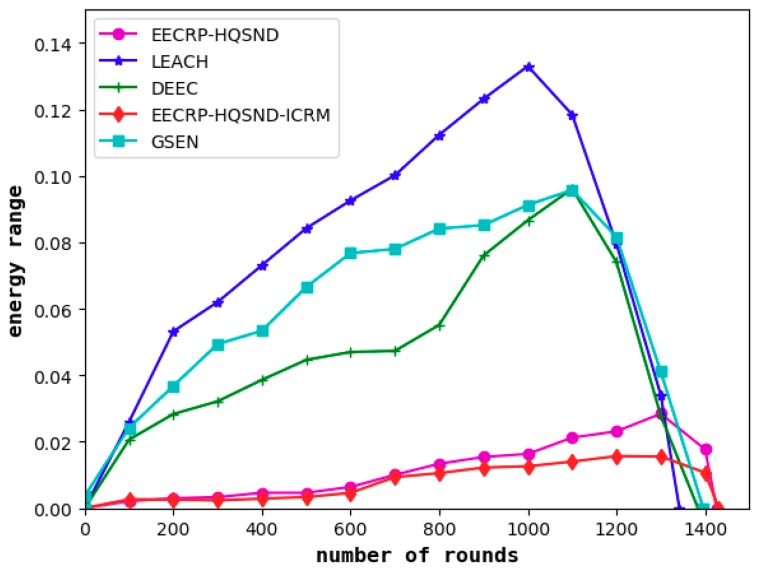
The range of the residual energy of the nodes in the network.

**Figure 21 sensors-19-02752-f021:**
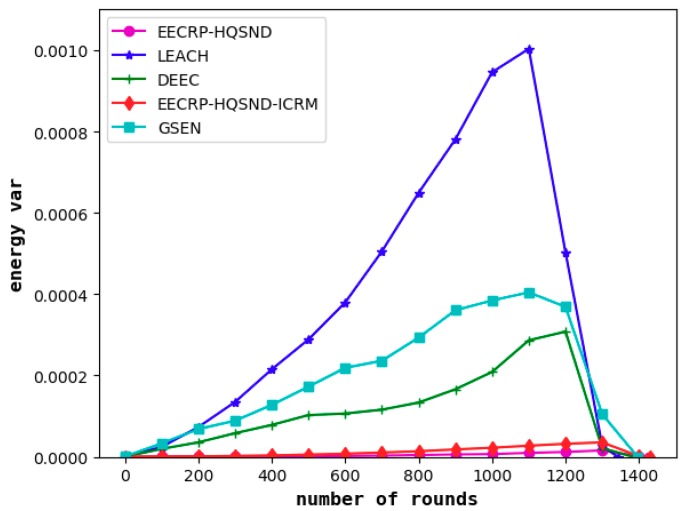
The variance of the residual energy of the nodes in the network.

**Figure 22 sensors-19-02752-f022:**
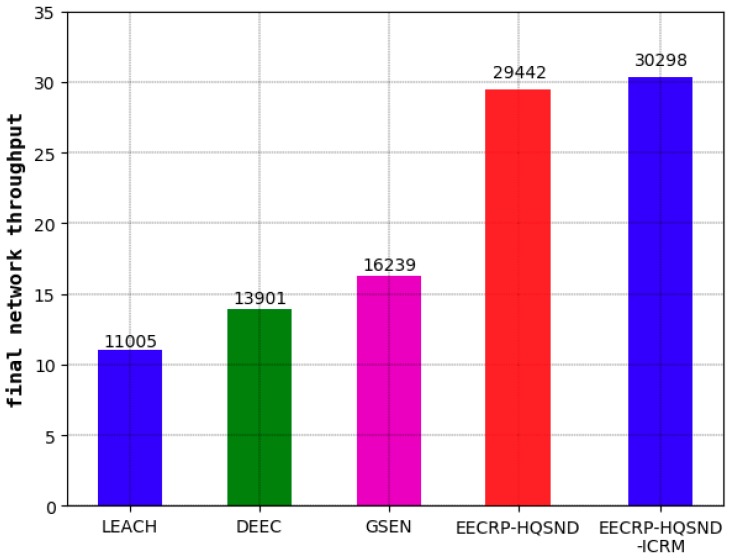
The comparison of the final throughput in the five protocols.

**Table 1 sensors-19-02752-t001:** Simulation parameters.

Parameter	Value
*E_elec_*	50 nJ/bit
*E_DA_*	5 nJ/bit/message
*ε_fs_*	10 pJ/bit/m^2^
*ε_mp_*	0.0013 pJ/bit/m^4^
The size of monitoring area	100 m*90 m
The coordinate of BS	(50, 45)
Initial number of nodes, *N*	91
Size of message, *l*	4000 bits
Initial energy	0.5 J
Perceptual radius, *R*	10 m

**Table 2 sensors-19-02752-t002:** Comparison of the quality of service (QoS) of the four node deployment strategies.

Strategy	Coverage	Integrity	Validity	Redundancy	Amount
Random	94.233%	86.689%	58.854%	41.146%	91
Uniform	100%	100%	63.134%	36.866%	110
1-Coverage Cellular Grid	100%	59.833%	99.889%	0.111%	46
Our Strategy	100%	100%	83.480%	16.520%	91

**Table 3 sensors-19-02752-t003:** Comparison of the number of rounds leading to decline in the network for each protocol.

Protocol	First Node Death	Coverage Degradation	Integrity Degradation
LEACH	1061	1093	1072
DEEC	1135	1175	1149
GSEN	1163	1227	1172
EECRP-HQSND	1331	1333	1332
EECRP-HQSND-ICRM	1364	1367	1365

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
