# Peer review of "An Energy-Efficient Clustering Routing Protocol Based on a High-QoS Node Deployment with an Inter-Cluster Routing Mechanism in WSNs"

_sensors, 2019, doi:10.3390/s19122752_

Reviewer 1 Report

The authors have proposed an energy efficient routing protocol for WSNs. There has been a lot of work done on designing energy efficient routing protocols for WSNs. Also, the authors have  recently published the following similar work:

 "An energy efficient Clustering routing protocol for wireless sensor networks based on AGNES with balanced energy consumption optimization".

However, the above work is not cited, which limits the contributions of this paper. Moreover, analysis of the routing protocols is missing only simulations are provided.

Author Response

Point 1: The authors have proposed an energy efficient routing protocol for WSNs. There has been a lot of work done on designing energy efficient routing protocols for WSNs. Also, the authors have recently published the following similar work: "An energy efficient Clustering routing protocol for wireless sensor networks based on AGNES with balanced energy consumption optimization". However, the above work is not cited, which limits the contributions of this paper.

Response 1: We are very grateful to your comments for the manuscript. This is a value comment. According to your suggestion, we have put the basic content of the above article in the introduction section and marked it in the reference. All of the modifications were highlighted in red.

Point 2: Moreover, analysis of the routing protocols is missing only simulations are provided.

Response 2: Thank you for your value comment. For the comment you have made, we have made two corresponding modifications in the paper:

       In line 245-249, we have analyzed in detail the advantage of selecting CHs in each cell.

       In line 337-341, we have analyzed in detail the disadvantage of applying the Dijkstra algorithm to this situation.

All of the modifications were highlighted in red.

So, the two analyses provide the basis for the relevant schemes adopted in this protocol. Then, the simulation results show us a more thorough analysis of the protocol by comparing the protocol in this paper with other protocols.

Thank you very much for your comments for the manuscript. We tried our best to improve the manuscript and made some changes in the manuscript. These changes will not influence the content and framework of the paper. We appreciate for your warm work earnestly, and hope that the correction will meet with approval. Once again, thank you very much for your comments and suggestions.

Reviewer 2 Report

This paper presents EECRP-HQSND-ICRM for WSNs.

The paper is well-written and research design is appropriate.

Here are three minor comments:

1) In equation 13, what is the 'k' and 'e'?

It is required to describe how the equation 13 could be derived.

2) In figure 21, why the EECRP-HQSND-ICRM is worse than EECRP-HQSND?

3) In figure 22, what does "network final througput"? It is required to define it clearly.

Author Response

Response to Reviewer 2 Comments

Point 1: In equation 13, what is the 'k' and 'e'? It is required to describe how the equation 13 could be derived.

Response 1: Thank you for your value comment. For the comment you have made, we have made some necessary changes in the equation. The energy consumed by the CH is divided into three parts: the energy consumed by receiving, the energy consumed by fusing, and the energy consumed by transmitting, the formulas of the above three are shown as 13, 14, 15.

And because in the last paper, we used ‘e’ to represent the size of message, but here we use ‘l’ to represent it, so the ‘e’ in the previous formula actually represents the size of message. We forgot to change it to ‘l’, now we have changed ‘e’ to ‘l’. Additionally, ‘k’ represents the number of CMs in the cluster.

All of the modifications were highlighted in red.

Point 2: In figure 21, why the EECRP-HQSND-ICRM is worse than EECRP-HQSND?

Response 2: Thank you for your value comment. I-CRM mainly achieves the reduction of CHs energy consumption by constructing the optimal transmission path of each CH and BS in the network, thereby prolonging the network lifetime. And from the previous figures, it can be found that compared with the protocol without the mechanism, the protocol with the mechanism can further reduce the network energy consumption and prolong the network lifetime.

       As for energy consumption balance, look as the following two tables.

       As can be seen from Table 1 and Figure 20, it is obvious that the I-CRM can reduce the range of the residual energy of the nodes. According to Table 2, it can be seen that the level of variance has been reduced to E-5, E-6 and E-7 compared to the E-2 and E-3 levels of range, so the increase in the variance is far less than the reduction in the range and the reduction in energy consumption.

Although I-CRM slightly increases the variance of the residual energy of the nodes during the network iteration, the increase in the variance can be said to be negligible compared to the reduction in the range and the reduction in energy consumption. Additionally, compared with LEACH, DEEC and GSEN, the variance of the residual energy in this protocol reaches approximate 0. Therefore, the protocol in this paper can balance energy consumption well.

On the other hand, the CHs selection mechanism through partition based on the uniformity of the CHs distribution in this paper can already do a good job of energy balance, and ICRM is only based on this, further reducing the energy consumption of nodes.

Point 3: In figure 22, what does "network final throughput"? It is required to define it clearly.

Response 3: Thank you for your value comment. As the nodes in the network die continuously, the number of CHs in the network decreases, and the success rate of transmitting packets to the BS also decreases. Therefore, the number of data packets transmitted in each round will gradually decrease. Until the last node dies, the network throughput reaches a maximum value, which is just the final network throughput. Now we have defined it in Section 4.7, which was highlighted in red.

 Thank you very much for your comments for the manuscript. We tried our best to improve the manuscript and made some changes in the manuscript. These changes will not influence the content and framework of the paper. We appreciate for your warm work earnestly, and hope that the correction will meet with approval. Once again, thank you very much for your comments and suggestions.

Reviewer 3 Report

The paper is focused on reducing and balancing the network energy consumption and extend the network lifetime. The topic itself is not novelty but the authors present a different proposal and make comparisons with other well-known solutions. 

The paper is well written: the introduction presents properly the problem to be solved, similar related works, the contribution and the structure of the paper. The proposal is well formulated and simulated. Feedback is presented at conclusions section and also the limitations of the proposal are reflected at the end of the paper. 

Author Response

Response to Reviewer 3 Comments

       At first, thank you very much for giving us such a high evaluation of our paper. In order to make the main theme of the paper clearer, we have made some modifications in the paper. All of the modifications have been marked in red, including: a reference to a paper we published earlier, a detailed description of the formula symbol, and a more detailed definition of the final network throughput, etc.

Thank you very much for your comments for the manuscript. We tried our best to improve the manuscript and made some changes in the manuscript. These changes will not influence the content and framework of the paper. We appreciate for your warm work earnestly, and hope that the correction will meet with approval. Once again, thank you very much for your comments and suggestions.

Round  2

Reviewer 1 Report

I have reviewed this paper before and my major concern was the contributions of the paper and missing analysis of the proposed technique. Both these issues still exist in the paper, the authors cited their paper now [12] after pointing out in the first review but they did not clarify that how current work has more contributions than [12]. Also, the analysis is still missing, there are merely basic energy consumption equations.

Author Response

Response to Reviewer 1 Comments

Point 1: The authors did not clarify that how current work has more contributions than [12].

Response 1: We are very grateful to your comments for the manuscript. This is a value comment.

In fact, the random node deployment strategy in [12] does not guarantee the obtained information’s QoS, thus must not meet the requirement of the QoS for the actual application. In addition, it does not consider the inter-cluster routing mechanism, which will cause some CHs to deplete energy prematurely because of the long transmission distance from the BS.

Based on these, this paper proposes a series of definition formulas for information integrity, validity, and redundancy, and a node deployment strategy based on 2-fold coverage. The simulation shows that the node deployment strategy in this paper is better than the random node deployment strategy in [12]. In addition, this paper optimizes the Dijkstra algorithm, including: (1) nonessential path neglecting considerations, and (2) a simultaneous introduction of end-to-end weights and path weights, achieving the construction of the inter-cluster routing mechanism (I-CRM). The simulation shows that the protocol with I-CRM can further extend the network lifetime compared with the protocol without I-CRM.

So, compared with [12], this paper has two more contributions: 1) A node deployment strategy can achieve better QoS performance; 2) The construction of the I-CRM can further extend the network lifetime.

Now, we have made modifications in line 107-111. All of the modifications were highlighted in red.

Point 2: Also, the analysis is still missing, there are merely basic energy consumption equations.

Response 2: Thank you for your value comment. For the comment you have made, combining with the simulation results, we have made corresponding analysis in the paper:

In line 425-438, combining with the alive nodes in the network with the iteration rounds, we have analyzed in detail the advantages of the CHs selection mechanism and ICRM.

       In line 450-453, combining with the network coverage in the network with the iteration rounds, we have analyzed in detail the advantages of the CHs selection mechanism and ICRM.

       In line 462-464, combining with the network integrity in the network with the iteration rounds, we have analyzed in detail the advantage of the ICRM.

In line 473-479, combining with the number of death rounds in the first node, the number of rounds leading to the network coverage’s decline and the information integrity’s decline, we have analyzed in detail the advantages of the CHs selection mechanism and ICRM.

       In line 549-557, we have analyzed in detail the advantages of the CHs selection mechanism and ICRM in Conclusions.

All of the modifications were highlighted in red.

Thank you very much for your comments for the manuscript. We tried our best to improve the manuscript and made some changes in the manuscript. These changes will not influence the content and framework of the paper. We appreciate for your warm work earnestly, and hope that the correction will meet with approval. Once again, thank you very much for your comments and suggestions.

Round  3

Reviewer 1 Report

Thanks for clarifying my comments. The contributions of the paper are now more clear and focused. One minor thing is that for CH selection, the authors consider both energy and distance to the BS and use fuzzification e.g.,  https://doi.org/10.3390/s16091459. This will improve further the cluster formation and CH selection mechanism.

Author Response

Response to Reviewer 1 Comments

Point 1: Thanks for clarifying my comments. The contributions of the paper are now more clear and focused. One minor thing is that for CH selection, the authors consider both energy and distance to the BS and use fuzzification e.g., https://doi.org/10.3390/s16091459. This will improve further the cluster formation and CH selection mechanism

Response 1: At first, thank you very much for you providing such a valuable paper.

We have carefully read it. The paper mainly uses fuzzification to obtain the cluster structure, and then considers each factor to find the optimal CH in each cluster. Undoubtedly, the paper and the paper we published before both adopt the strategy of first constructing clusters and then selecting CHs. Besides, it is also a feasible method to first select CHs and then construct clusters.

If we choose the method of first selecting CHs and then establishing clusters, we do not have to consider the distance between the CH and the member nodes when selecting CHs. So when building the cluster structure, it is natural to obtain the cluster structure in which the nodes are distributed closely to each other. Additionally, in order to distinguish this paper from the previous paper, we adopted the strategy of first selecting CHs and then establishing clusters.

According to the suggestion of the paper you provided, we have optimized the CHs selection mechanism, including: taking the factor (the distance from the node to the BS) into consideration. The simulation results show that the performance of the protocol is further improved. So, in the end, we would like to express our sincere thanks to you.

All of the modifications were highlighted in red.

Thank you very much for your comments for the manuscript. We tried our best to improve the manuscript and made some changes in the manuscript. These changes will not influence the content and framework of the paper. We appreciate for your warm work earnestly, and hope that the correction will meet with approval. Once again, thank you very much for your comments and suggestions.
